# Hierarchical association of COPD to principal genetic components of biological systems

Daniel E. Carlin[1], Simon J. Larsen[2], Vikram Sirupurapu[1], Michael H. Cho[3], Edwin K. Silverman[3], Jan Baumbach[4], Trey Ideker[1]*

1 Department of Medicine, Division of Genetics, University of California San Diego, La Jolla, CA, United States of America, 2 University of Southern Denmark, Odense, Denmark, 3 Channing Division of Network Medicine, Brigham and Women's Hospital, Boston, MA, United States of America, 4 Department of Computational Systems Biology, University of Hamburg, Hamburg, Germany

* tideker@ucsd.edu

**Data Availability Statement:** Data cannot be shared publicly because of patient identifiable information. Data are available through the COPD GOLD: https://goldcopd.org/ Data is also available

## Abstract

Many disease-causing genetic variants converge on common biological functions and pathways. Precisely how to incorporate pathway knowledge in genetic association studies is not yet clear, however. Previous approaches employ a two-step approach, in which a regular association test is first performed to identify variants associated with the disease phenotype, followed by a test for functional enrichment within the genes implicated by those variants. Here we introduce a concise one-step approach, Hierarchical Genetic Analysis (Higana), which directly computes phenotype associations against each function in the large hierarchy of biological functions documented by the Gene Ontology. Using this approach, we identify risk genes and functions for Chronic Obstructive Pulmonary Disease (COPD), highlighting microtubule transport, muscle adaptation, and nicotine receptor signaling pathways. Microtubule transport has not been previously linked to COPD, as it integrates genetic variants spread over numerous genes. All associations validate strongly in a second COPD cohort.

## Introduction

A longstanding challenge with Genome-Wide Association Studies (GWAS) is the so-called multiple testing problem that arises from testing thousands to millions of Single Nucleotide Polymorphisms (SNPs) for association with a phenotype (Bush and Moore 2012). Given a large number of association tests relative to the size of the population, a fraction of SNPs may appear to associate with the phenotype by chance. This problem is most commonly overcome by controlling the family-wise error rate, with $p \leq 5 \times 10^{-8}$ being the *de facto* significance threshold [1]. This strict significance threshold, in turn, has the consequence that strong effect sizes or very large population sizes, or both, can be necessary for causal genetic variants to be discovered.

A second, independent, challenge relates to functional interpretation. For any SNPs that meet the association threshold, the immediate task is to understand the molecular and cellular mechanisms that mediate the effects of those variants on phenotypic outcome. Linking SNPs to functions is complicated by multiple factors, including (1) the extensive SNP-SNP

through dbGaP: https://www.ncbi.nlm.nih.gov/projects/gap/cgi-bin/study.cgi?study_id=phs000179.v6.p2 All COPD Gene genotype and phenotype files exist on dbGaP with accession number phs000765. COPDGene data are available through monitored public access at dbGaP. This is an NIH repository for data, and both phenotype and genotype data from COPDGene are deposited there. Since the genotype data are identifiable, COPDGene was designed for monitored public access, not unmonitored public access. Data access requests are dealt with solely by dbGaP staff. The COPDGene investigators do not have any control over who receives access to the data on dbGaP. All 21 IRBs at the COPDGene clinical centers approved the monitored public access approach for COPDGene data. The consent form for COPDGene indicated that subject data would only be made available through monitored public access.

**Funding:** SL, DC, and TI were supported by grants from the National Institutes of Health (NIH) to TI, R01 HG009979, P41 GM103504 and P50 DA037844. ES, MC and the COPDGene project described were also NIH-supported under grants U01 HL089897 and U01 HL089856. JB and SL received financial support from VILLUM Young Investigator grant 13154 to JB as well as H2020 grants RepoTrial 777111 and FeatureCloud 826078. The funders had no role in study design, data collection and analysis, decision to publish, or preparation of the manuscript.

**Competing interests:** TI is co-founder of Data4Cure, Inc., is on the Scientific Advisory Board, and has an equity interest. TI is on the Scientific Advisory Board of Ideaya BioSciences, Inc., has an equity interest, and receives sponsored research funding. The terms of these arrangements have been reviewed and approved by the University of California San Diego in accordance with its conflict of interest policies. MC is a consultant for Illumina and AstraZeneca. In the past three years, EKS has received institutional grant support from GSK and Bayer. This does not alter our adherence to PLOS ONE policies on sharing data and materials.

covariation observed at genomic loci due to linkage disequilibrium (LD) [2]; (2) the mapping of SNPs to genes, since an associated locus can either lie far from gene bodies or encompass many; and (3) the fact that complex diseases arise not from the action of a single gene but from the integrated effects of diverse genetic loci acting on common or opposing functions [3].

A body of previous work has attempted to tackle these challenges by moving from tests of individual SNPs to tests based on genes and gene sets. This field, also known as pathway or network GWAS, has given rise to many methods [4–16] and has been extensively reviewed [17–27]. The benefits of these approaches are twofold. Firstly, the number of genes and functional gene sets is substantially fewer than the number of SNPs, yielding a reduced set of hypothesis tests with the potential to discover variants with weaker effects than would be detected when considering SNPs individually. Secondly, these approaches allow for functional interpretation of GWAS results by associating phenotypes with sets of genes operating in common biological pathways. Such association can point to underlying mechanisms that would not be revealed otherwise [18].

Among these methods, a general template is the "two-step" test, in which a standard single-variant association test is first performed to obtain a summary statistic (e.g. $p$-value of association) for each SNP. From these SNPs, a short list of complementary representatives is selected (e.g. without strong covariation, $r^2 < 0.5$). Then, in the second step, each gene and functional gene set is assigned a score equal to the mean test statistic among the SNP representatives, and this score is tested for significance using permutations. Versions of this approach are offered by standard tools such as PLINK [4,8], which provides a self-contained method for testing not only single SNPs, but also sets of SNPs and sets of genes, for association. Several newer alternatives, including MAGMA [9], implement a gene set test using a linear regression framework. MAGMA first performs principal component regression (PCR) of the phenotype against the SNPs in the neighborhood of each gene, yielding a per-gene significance of association (p-value). Each gene set of interest is then tested by examining the distribution of per-gene scores that make up that gene set, in comparison to a null distribution. Other notable gene set tests include ALIGATOR [5], MAGENTA [6], INRICH [7], GSA-SNP2 [16], and Generalized Berk-Jones [14].

These methods all share the intuition that knowledge of gene function can increase statistical power and interpretability of GWAS, by pooling signals across sets of genes organized by common functions. Thus far, however, the essential unit of genome-wide association has remained the SNP or the gene. A standard GWAS is still run to compute SNP level summary statistics over the study population, after which information from individual genotypes is not used further; rather, the $p$-values of association of SNPs within each gene set are evaluated for unexpected distributions.

Here, we extend the intuition of functional GWAS by treating the function itself as a basic genetic unit, rather than the SNP or gene. In contrast to the previous two-step structure which computes association statistics at the SNP level before proceeding to analyze gene sets, we develop an approach in which the principal genetic effects on each function are captured and made subject to direct tests of association in human populations. A direct "one-step" test has the potential to detect function-phenotype associations based on the convergent effects of many SNPs, even when the marginal effects of each of these SNPs may be insignificant. It also sidesteps some of the previous statistical challenges that have been raised with the two-step procedure [16], such as the double-counting of SNP contributions near genes that are members of multiple pathways, or inconsistent treatment of genes of different lengths and local LD structures. The general problem definition is to seek a low dimensional representation of SNPs covered by each known function and to identify which of these functions are associated with phenotype using regression models (Fig 1). Since biological functions are organized

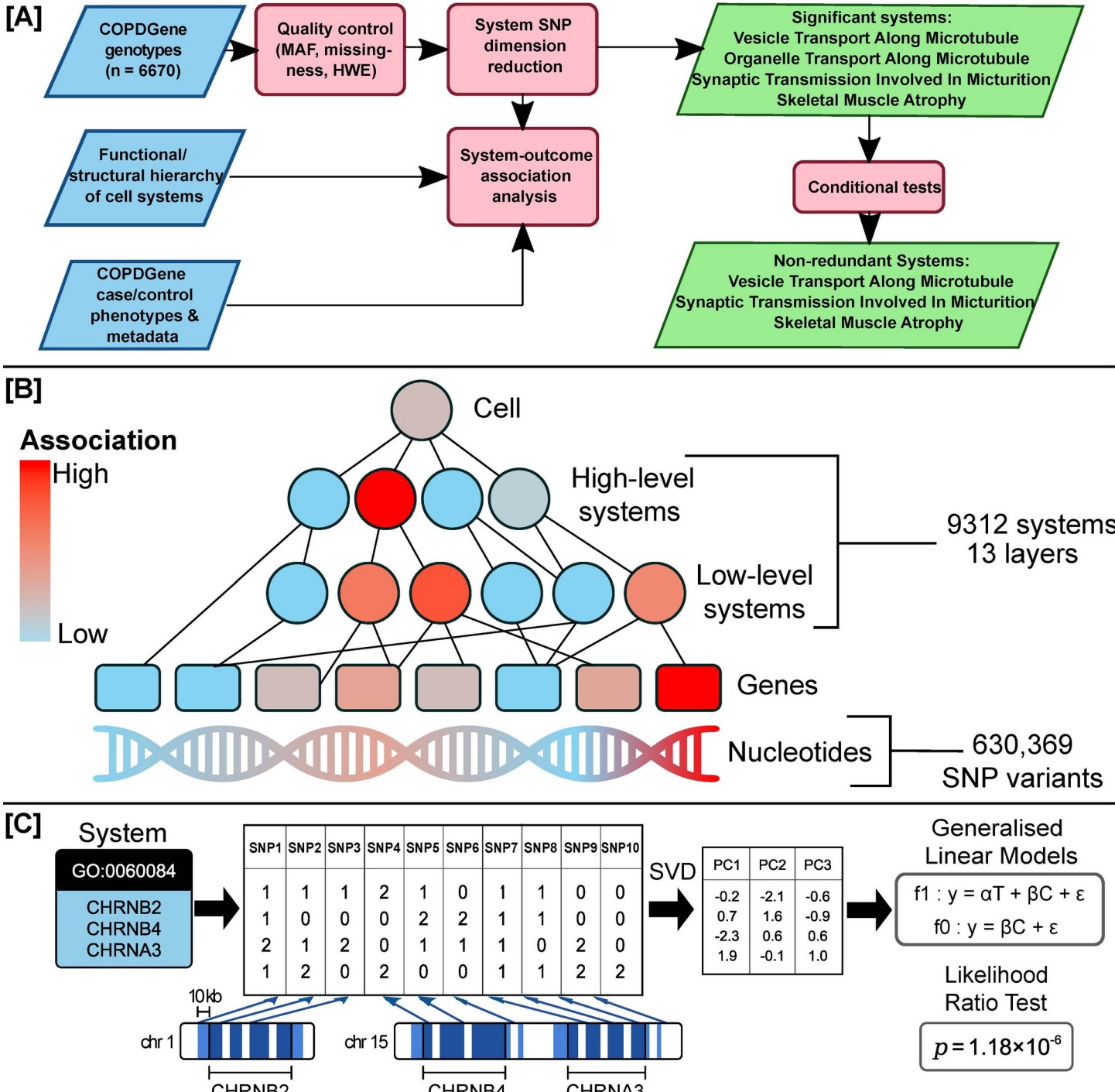

**Fig 1. Study design.** (A) Analysis workflow. Blue rhomboids are data inputs and green rhomboids are analysis outputs. MAF, Minor Allele Frequency. HWE, Hardy-Weinberg Equilibrium. (B) Hierarchy of functions increasing in specificity from top to bottom. Color shows strength of association. (C) Dimensionality reduction and association workflow. For each system defining a set of genes (left), a SNP matrix (center) is constructed from variants near system genes (blue bands in underlying chromosomes). Matrix columns: SNPs; Rows: Individuals. Values (0, 1, 2) are (major allele homozygous, heterozygous, minor allele homozygous) at each SNP. A low-dimensional representation is computed using SVD (center right) and tested for association with phenotype (model f1) in comparison to the null hypothesis (f0).

hierarchically, from specific to general resolutions, a follow-up panel of statistical tests is applied to identify which of these resolutions yields the most plausible association. We show that, despite its conceptual simplicity, the method identifies pathway associations that have not been previously reported in conventional GWAS or in gene set analyses. The suggested method, which we call Hierarchical Genetic Association Analysis (Higana), is publicly available as a Python implementation on GitHub at http://github.com/SimonLarsen/higana/.

This type of analysis is explored in the context of Chronic Obstructive Pulmonary Disease (COPD), a progressive lung disease characterized by respiratory symptoms (including shortness of breath) and airflow obstruction. COPD has an estimated 384 million cases worldwide [28] and three million deaths annually [29], making it one of the leading causes of death. Tobacco smoking is the most significant risk factor although other factors have been implicated as well, such as childhood asthma, air pollution, and occupational exposure to dusts and chemical fumes [30]. COPD also involves the combination of these environmental factors with genetics, with a study of Danish and Swedish twins suggesting that as much as 60% of disease susceptibility can be explained by the additive effects of genetic variants [31]. In an effort to systematically elucidate genetic risk factors, COPDGene, a multicenter study, has built a large cohort of COPD cases and controls with phenotypes that include chest CT scans and assessment of emphysema, gas trapping, and airway wall thickening [32]. Very recently this cohort has been analyzed in conjunction with more than 20 other studies to identify 82 genome-wide significant variants associated with COPD, with these variants explaining approximately 7% of the variance in phenotype [33].

## Results

### Hierarchical genetic analysis of COPD identifies systems of convergent SNPs

We applied hierarchical genetic association analysis to the non-Hispanic White COPDGene cohort [32] of 2812 cases and 2534 controls (Fig 1, Methods). A hierarchy of 9312 systems was selected from the GO knowledgebase [34], and each of these systems was attached to a data matrix providing, for each individual, measurements of SNPs proximal to the genes in that system. Each matrix was analyzed to extract its principal components, representing a simple compact representation of the genetic state of that system across the samples (i.e. *system genetic components*, Methods). We then applied a regression framework, using standard statistical approaches, to test the association of system genetic components with the COPD phenotype while accounting for general genetic and clinical covariates.

Among the 9312 tested systems, our analysis identified four for which the components were significantly associated with COPD at a strict significance threshold controlling for the Family-Wise Error Rate (FWER < 0.05, adjusting the association p-value using a Bonferroni procedure, Table 1 and Fig 2, complete results S1 Data). We noted that each of these systems maintained its significant phenotype association when performing a competitive association test against the equivalent system in a 'null' ontology hierarchy for which the gene-to-ontology annotations had been randomized (Methods, Table 1 'ontology permutation' column). All significantly associated systems were from the Biological Process domain of GO. The first two of these, *Organelle Transport Along Microtubule* (GO:0072384, FWER < 0.00127) and *Vesicle Transport Along Microtubule* (GO:0047496, FWER < 0.00954), contained dozens of genes and appeared to represent a convergence of signal from multiple loci dispersed throughout the genome. Notably, none of these systems contained any genome-wide significant SNPs ($p < 5 \times 10^{-8}$) nor SNPs in LD with genome-wide significant SNPs (none with $r^2 > 0.2$ within 1 kb). In particular, *Vesicle Transport Along Microtubule* aggregated the individual effects of SNPs in

**Table 1. Top ten systems most significantly associated with COPD status.**

| System | GO Branch | Genes | p-value | p-value Bonferroni adjusted | p-value in validation set | p-value in ontology permutation test |
|---|---|---|---|---|---|---|
| Vesicle Transport Along Microtubule (GO:0047496) | BP | 37 | 1.36E-07 | 0.001 | 0.001 | 0.01 |
| Organelle Transport Along Microtubule (GO:0072384) | BP | 58 | 1.02E-06 | 0.01 | 0.373 | 0.01 |
| Skeletal Muscle Atrophy (GO:0014732) | BP | 3 | 1.12E-06 | 0.01 | 0.026 | 0.01 |
| Synaptic Transmission Involved in Micturition (GO:0060084) | BP | 3 | 1.18E-06 | 0.011 | 0.013 | 0.01 |
| Striated Muscle Atrophy (GO:0014891) | BP | 5 | 2.89E-05 | 0.269 | 0.826 | Not tested |
| Muscle Atrophy (GO:0014889) | BP | 6 | 4.69E-05 | 0.437 | 0.838 | Not tested |
| Regulation of Erythrocyte Differentiation (GO:0045646) | BP | 33 | 6.65E-05 | 0.619 | 0.537 | Not tested |
| Paranode Region of Axon (GO:0033270) | CC | 6 | 1.00E-04 | 0.931 | 0.574 | Not tested |
| Chemorepellent Activity (GO:0045499) | MF | 22 | 1.16E-04 | 1 | 0.863 | Not tested |
| Skeletal Muscle Adaptation (GO:0043501) | BP | 6 | 1.24E-04 | 1 | 0.757 | Not tested |

or near genes including *AP3S1* on chromosome 5, *DYNC1I1* on chromosome 7, *KIF5A* on chromosome 12, *CLN3* on chromosome 16, and *KIF3B* on chromosome 20 (Fig 3). This system, considered as a whole, represented greater genetic signal than any of the supporting SNPs, which individually had not been implicated in previous GWAS analysis [32,33].

The third system associated with COPD contained three genes involved in *Skeletal Muscle Atrophy* (GO:0014732, FWER < 0.01047). One of these, *IREB2*, was at the 15q25.1 locus with SNPs that were also genome-wide significant when considered individually (S2 Fig). The protein encoded by *IREB2* is an RNA-binding protein (IRP2) involved in maintaining iron homeostasis, and it had previously been associated with COPD and lung function in multiple studies including GWAS, analysis of mRNA expression and murine functional studies [35–38]. The fourth associated system was *Synaptic Transmission Involved in Micturition* (GO:006008, FWER < 0.01098). This system contained three genes, *CHRNB2*, *CHRNB3*, and *CHRNB4*, encoding neuronal acetylcholine receptor (nAChR) subunits involved in the response to acetylcholine neurotransmitters and nicotine [39]. One of these nAChR genes, *CHRNB4*, was at the same genome-wide significant locus, 15q25.1, as was the aforementioned *IREB2*. Looking beyond the four reported systems using a relaxed threshold of significance, we noticed that among the ten most associated systems, five contained genes near 15q25.1. Further inspection suggested that the COPD association scores of all of these systems were predominantly due to this single genetic locus. Indeed, 15q.25.1 was the first locus to be implicated in a COPD GWAS [35] and has been replicated [40–42].

## SNPs converge specifically on transport of vesicles and not other organelles

We noted that the system *Vesicle Transport Along Microtubule* (GO:0047496) was a child ("is a" relationship) of *Organelle Transport Along Microtubule* (GO:0072384). Since both systems were significantly associated with COPD (Fig 2), we considered that these might be redundant results based on the same underlying constellation of SNPs. Alternatively, it was possible that

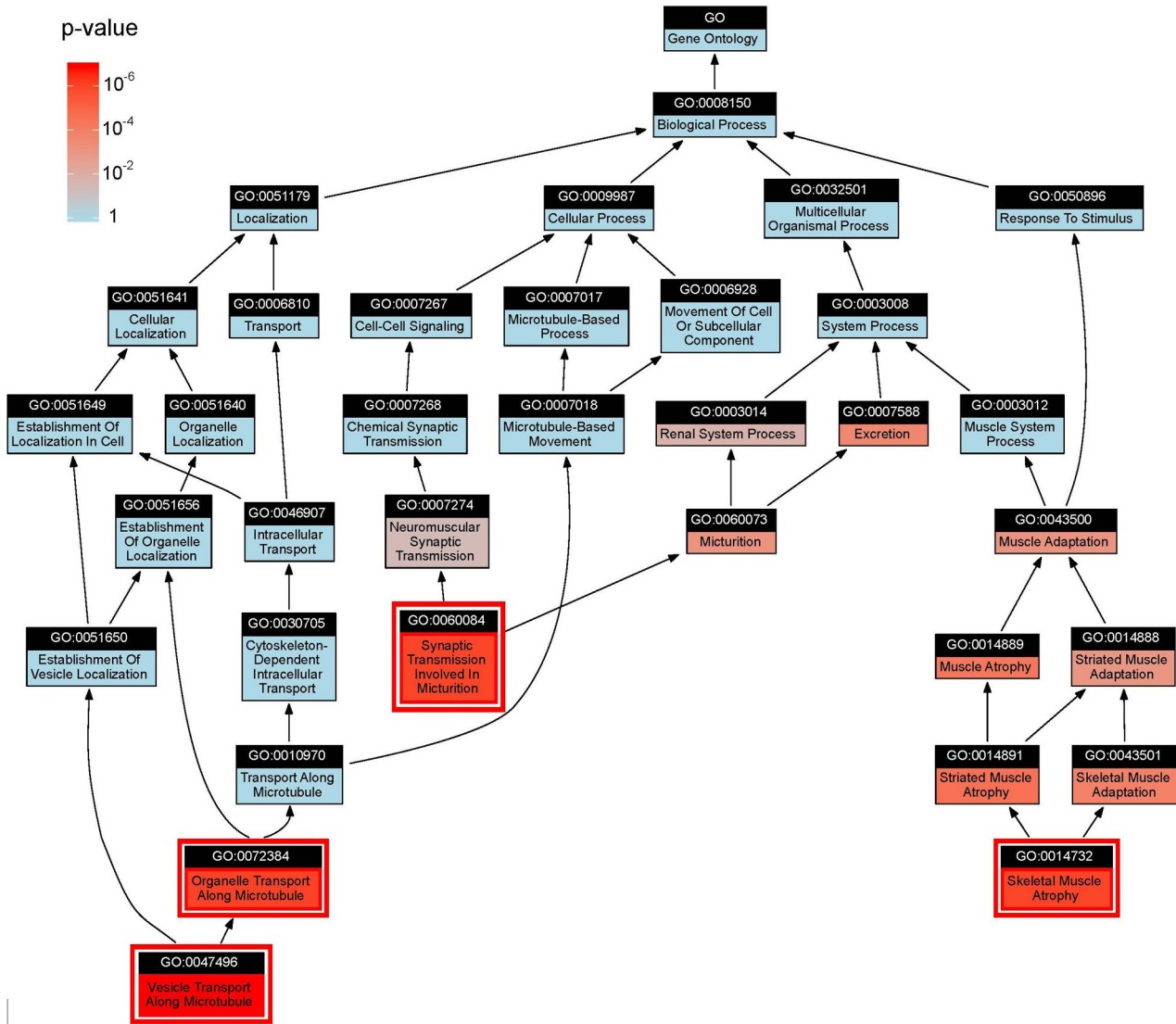

**Fig 2. Subhierarchy of significantly associated systems.** System color corresponds to unadjusted p-value. Systems with FWER < 0.05 are highlighted with red outline.

the parent system integrated SNPs in the child with SNPs in yet other genes or systems in determining an overall effect on phenotype. To distinguish between these two alternatives, we implemented a pair of additional association tests to evaluate the phenotype association of each system in the context of its most associated child system or annotated gene, respectively (Methods). In particular, the "Top child test" removed all SNPs assigned to the most associated child system (if any) and repeated the association analysis in this reduced genetic context. The "Top gene test" did the same but for the most associated gene. These additional association tests were run on all of the ten most significantly associated systems (Fig 4).

When disregarding its top child, the parent system *Organelle Transport Along Microtubule* lost all association with COPD. In contrast, the child, *Vesicle Transport Along Microtubule*, lost very little signal when disregarding its own four children or its top gene. Moreover, these four children exhibited relatively low COPD association when considered individually (Fig 4B).

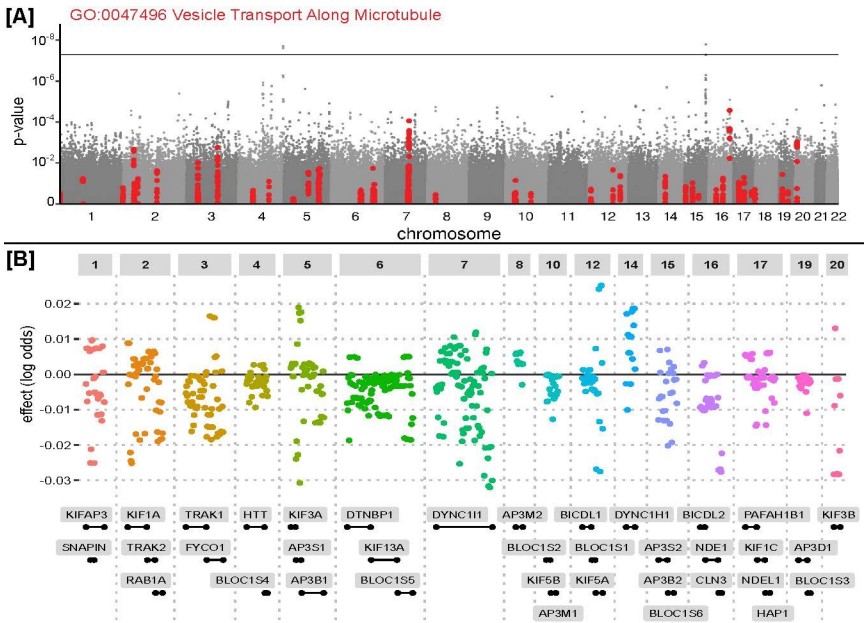

**Fig 3. Association results for the *Vesicle Transport Along Microtubule* system (GO:0047496).** (A) Manhattan plot of the standard GWAS results for the primary COPD cohort. SNPs assigned to genes associated with GO:0047496 are highlighted in red. Horizontal line marks the standard GWAS significance threshold, $p = 5 \times 10^{-8}$. (B) Contributed effects of SNPs in the principal component regression model (system genetic components) for GO:0047496. SNPs are sorted by genomic position. Horizontal distance between SNPs does not correspond to genomic distance. SNP effect is the change in log odds ratio per minor allele copy contributed to the principal component regression for the entire system.

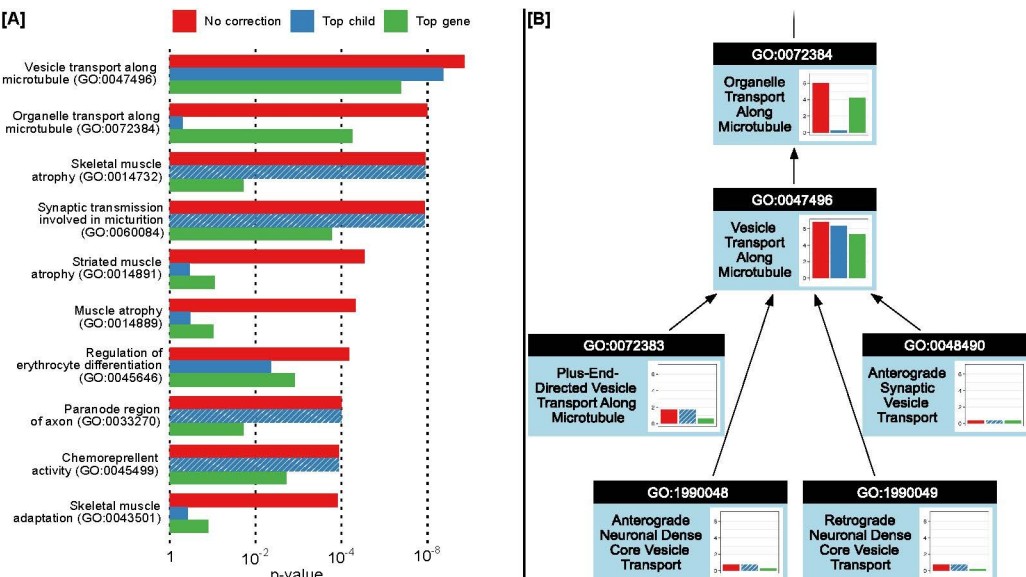

**Fig 4. Conditional association test results.** (A) Comparison of association results for the original uncorrected analysis (red) and conditional tests (blue, green) for the top 10 most associated systems. (B) Conditional test results for the vesicle transport subhierarchy. P-values are unadjusted. Shaded bars for the "top child" test indicate the system had no children and thus no correction was applied.

These results indicated that *Vesicle Transport Along Microtubule* was the appropriate resolution (i.e. level of specificity) at which SNPs at different genetic loci converge to affect COPD.

For other systems among the top ten, the additional significance tests showed that some were indeed better explained by a single significant child system or gene. For example, the group of related systems including *Skeletal Muscle Atrophy*, *Muscle Atrophy*, and *Skeletal Muscle Adaptation* appeared to be predominantly driven by the single gene *IREB2* (although for *Skeletal Muscle Atrophy* perhaps not entirely, S3 Fig). In other cases such as *Synaptic Transmission Involved in Micturition*, the genetic signal was more clearly preserved across the child and top-gene tests, supporting a specific COPD association at the resolution of the entire system.

**Comparison to other genetic analysis methods using gene sets.** We also evaluated the results we had obtained with the Higana approach against the gene set association tests implemented in PLINK [8] or MAGMA [9]. At the same 0.05 FWER cutoff used thus far, MAGMA reported no systems (S3 Data), and PLINK reported a single system (GO:0035094; *Response to Nicotine*, S4 Data), versus the four significant systems that had been reported by Higana (Table 2). At a more relaxed significance threshold, PLINK and MAGMA reported a larger number of systems. For example, at $p < 0.001$ (uncorrected for multiple hypothesis testing) PLINK reported 73 systems, versus 48 and 16 for Higana and MAGMA, respectively. However, given the number of tested systems the corresponding FWER was greater than 1, indicating a very high rate of false positive (type I) errors at this relaxed significance threshold.

We also evaluated the congruence of systems reported by each method and found that the results were largely complementary (S4 Fig). No systems were reported in common by PLINK and Higana at the corrected FWER < 0.05 cutoff, while 12 common systems were reported at the relaxed $p < 0.001$, representing 16% and 25% of systems reported by PLINK and Higana, respectively. The top two systems identified by Higana, *Vesicle Transport Along Microtubule* and *Organelle Transport Along Microtubule*, were not identified by other methods even at the relaxed cutoff. The *Response to Nicotine* system identified by PLINK was partially overlapping with the *Synaptic Transmission Involved in Micturition* system identified by Higana, both containing nAChRs (*CHRNB2*, *CHRNB3*, and *CHRNB4*). Comparing the complete set of system association outcomes for each method (i.e. the complete vector of association p-values over all systems), we observed moderate agreement between Higana and MAGMA (Pearson $\rho = 0.56$) and between Higana and PLINK ($\rho = 0.54$) with less agreement between PLINK and MAGMA ($\rho = 0.36$).

We found that Higana required significantly less computation time than PLINK but significantly more computation time than MAGMA. Further inspection suggested that the long run-time of PLINK is due to the large number of permutations needed to achieve accurate estimates of significance. The majority of the computation time needed by Higana is devoted to singular value decomposition of SNPs to formulate system genetic components. We found that this decomposition could be made significantly more efficient by using the approximate method of randomized matrix decomposition, at the cost of a small decrease in accuracy (Methods).

**Validation with a second independent cohort.** The COPD Gene study also interrogated an independent African-American validation cohort for COPD (821 cases, 1749 controls). We thus examined this cohort separately to validate our results, defining cases and controls in the same manner as the non-Hispanic white cohort and using the same covariates and GO structure. Higana results on this cohort confirmed three out of four of the globally significant systems (*Vesicle Transport Along Microtubule*, *Skeletal Muscle Atrophy*, and *Synaptic Transmission Involved in Micturition*) as also significant in the African American cohort (Table 1 and S5 Fig). The globally significant system that did not validate was *Organelle*

**Table 2. Number of significant systems by each method for different significance criteria.**

| Method | p<5e-6, Bonferroni adjusted p<0.05 | p<0.001 | Computation time |
|---|---|---|---|
| Higana | 4 | 48 | 2 hours, 32 minutes |
| PLINK | 1 | 73 | 5 days 16 hours |
| MAGMA | 0 | 16 | 2 minutes, 29 seconds |

*Transport Along Microtubule*, which had been discarded earlier as the redundant parent term of *Vesicle Transport Along Microtubule*.

Finally, we asked whether there was significant overlap in borderline significant systems with an association p-value < 0.1. A total of 2017 systems in the non-White Hispanic cohort and 1345 systems from the validation cohort met this criterion. Of these, 232 systems were in agreement between the two datasets, representing significant overlap (hypergeometric p = 8.28 $\times 10^{-6}$). This overlap suggests that relaxing the significance of association may indeed be a viable strategy for nominating additional functions associated with a disease phenotype.

**Simulation results.** We tested the difference in performance between a "one-step" one step statistical procedure such as the one that Higana implements and "two-step" procedure as employed by PLINK/Magma. In two simulation schemas, one where the SNPs were independently simulated and another where SNPs were simulated with an Ising model the one step simulation consistently was more powerful than the two step approach across all values of noise (S7 Fig).

## Discussion

We have described a simple yet powerful genetic analysis framework based on hierarchical knowledge of biological systems, in which genetic components of each system are subject to direct and conditional tests for associations with phenotype. When applied to a COPD case control study, the approach reports four systems as significantly associated with disease cases. Two of these systems related to microtubule transport have not been implicated previously in COPD, as individual genetic variants in these systems have levels of significance that are sometimes impressive ($p \approx 5 \times 10^{-5}$, Fig 3A) but do not beat the strict genome-wide significance threshold of a conventional GWAS in the COPD populations and population sizes examined thus far.

Subsequent conditional tests show that the responsible SNPs are contained completely in the smaller of the two microtubule systems, *Vesicular Transport Along Microtubules*, which nests hierarchically within the larger *Organelle Transport Along Microtubules*. Within vesicular transport the genetic signal is spread over many loci and genes, with the strongest association observed in the gene encoding cytoplasmic dynein 1 intermediate chain 1 (*DYNC1I1*), part of the cytoplasmic dynein 1 complex responsible for moving various cargoes along the cytoskeleton (Fig 3B). A recent analysis of the UK Biobank population has found that an intron variant in this gene, rs6961619, associates with another pulmonary trait, forced vital capacity (FVC) [43]. On the other hand, COPD depends not so much on FVC as forced expiratory volume (FEV1) and the FEV1 / FVC ratio.

In the case of the third significant system, *Skeletal Muscle Atrophy*, conditional testing determined that a single gene encoding an iron response element, *IREB2*, was responsible for most of the association, although other genes in this system, such as *MYOG* and *ACTN3*, may play a minor role. While the link between *IREB2* and pulmonary disease is still unclear, we see at least three potential mechanisms which are not mutually exclusive. Firstly, cigarette smoking has been associated with higher levels of iron in the lungs which may, in turn, contribute to

oxidative injury and pulmonary obstruction [44,45]. Secondly, skeletal muscle dysfunction is a common comorbidity of COPD [46] suggesting that genetic variation impacting this system may provide a common cause for the two conditions. Thirdly, the association of *Skeletal Muscle Atrophy* with COPD could be explained not by *IREB2* but by linkage disequilibrium with SNPs near *CHRNA3* and *CHRNA5* [42], encoding nicotinic acetylcholine receptors that respond to the acetylcholine neurotransmitter as well as nicotine [47]. These nAChRs have been linked to nicotine dependence in genetic association studies [48–51] and this locus has also previously been associated with COPD [35,41]. Overall, the locus containing IREB2 and nAChRs is a complicated genomic region, with mediation analysis suggesting the likely presence of two separate signals [52].

The nAChR gene family more clearly explains the significance of the fourth system, *Synaptic Transmission Involved in Micturition*, which on further inspection reduces to yet another three nAChR genes, the beta receptor subunits *CHRNB2*, *CHRNB3*, and *CHRNB4*. These genes arise here in the context of a micturition pathway, as nicotinic acetylcholine receptors are involved in autonomic control of urinary bladder [37]. However, we have no reason to expect that micturition and lung function are related other than through the pleiotropic functions of the *CHRNB* genes. This anecdote argues that the label given a system by a knowledge-base like GO can be misleading, since that label is assigned without disease context. Persisting despite the label, we see that the set of genes in the system corresponds to a rational nicotine receptor family on which genetic variants converge in COPD. It is perhaps surprising that functions related to nicotine response emerge from the analysis, since an individual's current smoking status and total lifetime pack-years were included as covariates in the hierarchical genetic association procedure (Methods). The fact that we still observe such systems in the significant results suggests that the impact of smoking on COPD incidence is more complex than is modeled by these two standard covariates.

When comparing Higana to PLINK and MAGMA in genetic analysis of COPD, we find that Higana and PLINK have greater power than MAGMA (Table 2). This finding contradicts the evaluation in the original MAGMA manuscript [9], where MAGMA was found to have more power than PLINK in a Crohn's disease cohort. This discrepancy may be due to a difference in population sizes, effect sizes, or the sizes and numbers of tested gene sets. The increased sensitivity of Higana may be attributed to the detection of multi-locus effects that become apparent only when the principal components of genotypes are calculated at the systems level. For example, system components can represent the cumulative burden of genetic variants on the collection of genes in a system, even when none of those variants individually show significant (if any) differences in incidence between cases and controls. Such cumulative genetic effects arise among pathways under selection for somatic mutations in cancer [53], and they may play a similar role in a germline context. Furthermore, since system components are regressed using a logistic regression (logit) function, not only can the total number of variants be counted (per patient per system) but AND/OR logic functions can be learned as well, e.g. a system that transmits its phenotypic effect given the right combination of haplotypes at two loci A (AND) B.

We note that in both of our simulation schemas, both independent and Ising model generated, the one-step Higana approach outperformed the two-step approach. The Fisher's p-value combination methods are known to have difficulty when hypotheses are not independent could explain the performance in the Ising model case. However, the one-step procedure outperforms in the independent case as well. This suggests that the combination into principal components versus the averaging-type procedure of the log probabilities serves to denoise the signal more effectively. Furthermore, we suggest that the collapse of different numbers of correlated variables from LD-blocks of various sizes, as well as variable gene set sizes to a limited number of

principle component variables decreases the heterogeneity of the statistical test, allowing for a more uniform application of the method to widely variable numbers of input SNPs.

Currently in the GWAS community there is a debate about the correct null model for testing the significance of association of a gene set. MAGMA utilizes a "competitive" null model, which states that the genes in the set are no more associated with the phenotype than other genes, accounting for variable heritability of phenotypes. However, such a model appears to come at the price of power in comparison to "self-contained" null models that seek to disprove the null hypothesis that no SNPs (or genes) are associated with the phenotype after accounting for all confounders [14]. Higana represents a unique entry in this debate, as it does not rely on gene-level tests for its initial identification of candidate gene sets; thus it avoids both power loss and anomalies due to genomic proximity of genes assigned to the same pathways [26].

We note that computing PCs for each gene set may have the effect of differentially weighting the contribution of different genes to the overall gene-set p-value. Such differential weight is a feature of the approach–i.e. to capture and properly weight the true causal variants within the gene set. On the other hand, this approach could also assign genes more weight if they have denser LD. Such bias is alleviated by the top gene test, which rejects gene sets driven by a single very dominant gene. Nonetheless, an interesting question for exploration in a future study is whether an additional correction for LD density across genes is desirable.

In computing the genetic components for each system, we adopted an unsupervised approach using Principal Component Analysis, rather than an approach supervised by the phenotype case/control label. An unsupervised approach makes it straightforward to adjust for confounding variables and to avoid the use of permutation tests for estimating significance. This design also allows later testing for association against many different phenotypes without the necessity to recompute the many principal components of each system, an important capability in analysis of cohorts with many measured phenotypes such as the UKBioBank cohort [54]. Conversely, a disadvantage is that one is unlikely to discover associations with large systems covering many genes, since the unsupervised projection preserves the dimensions with the highest variation over genotype, not the highest explanatory power over phenotype. The larger the set of genes, the more closely its projection resembles that of the genome-wide set of SNPs used to estimate (and correct for) the overall population structure. The use of principal component analysis may also favor SNPs with greater minor allele frequencies insofar as they contribute greater variance to the data matrix. Accordingly, another compelling future direction may be to explore supervised projection of data in formulating the genetic components of a system, perhaps recursively by examining the components of the system's children and/or parents. Alternatively, we note that Principal Component Analysis is but one choice for an unsupervised projection of a data matrix, and it is likely worthwhile to explore other unsupervised projections in future work.

While GO provides a broad model of cellular processes, it misses important associations due to incomplete knowledge of the cell and tissue biology of the disease, as well as poor or missing function definitions or gene annotations. Indeed, 20% of all human genes were not annotated to a system with 100 or fewer genes (S6 Fig) and thus were excluded from our analysis. Such omission is evidenced by the lack of COPD-associated systems including *FAM13A* on chromosome 4, despite there being two genome-wide significant SNPs within 10 kb of this gene [55] and supporting functional studies [56,57]. Increasing the maximum gene set size used for drawing systems from GO would be one way to address this problem (Methods). A preferable longer-term solution would be to identify and model the hierarchy of biological systems directly from primary experimental datasets, designed to elucidate disease pathways in highly relevant biological conditions and environmental contexts.

## Materials and methods

### Ethics statement

The COPDGene study protocol was approved by the institutional review boards at the 21 participating clinical centers. All COPDGene participants provided written informed consent.

### Experimental design

**COPD cohort and data.**   We obtained genotype and phenotype data from 6670 non-Hispanic white subjects enrolled in the COPDGene study [32]. All subjects were long-time smokers of at least 10 pack-years with varying degrees of COPD assigned according to the Global Initiative for Chronic Obstructive Lung Disease (GOLD) stages [58] Case individuals were determined as those with GOLD grades 2–4 (moderate to very severe COPD, n = 2812), while control individuals were defined as smokers with normal spirometry (n = 2534). Individuals that did not fit these case/control categories were excluded from further analysis. Genotypes were collected for each study participant using a custom SNP array, which after filtering and quality control resulted in a total of 630,369 distinct SNP variants. We refer to [32] for a detailed description of the study design and data collection methodology.

**Defining the functional hierarchy.**   We adopted the hierarchy of human biological systems provided by the Gene Ontology (GO) knowledgebase (The Gene Ontology Consortium 2018), which included 16,451 systems, known as GO terms, at time of download (release 2019-01-01). Systems were interconnected using "is a" and "part of" relations documented in GO, where each of these relations connects a more general "parent" to a more specific "child". The three aspects of GO (Molecular Function, Cellular Component and Biological Process) were combined under a common root to obtain a single, unified hierarchy. Sets of genes annotated to each system were also obtained from GO, using all annotation evidence codes except "inferred from electronic annotation" (IEA). Gene annotations were propagated through the hierarchy recursively, such that each system inherits all annotations of its children. After this operation, the set of genes assigned to each system was the union of the genes annotated to that system by GO and all genes annotated to any of its descendants. All systems with fewer than 3 or greater than 75 genes were removed, wishing to exclude systems too small to distinguish from a gene-level analysis and systems too general to functionally interpret. Furthermore, we removed systems for which the gene set was very similar to that of a child system. Specifically, we first defined the degree of parent-child similarity as $S(P, C) = |A_C| / |A_P|$, where $A_P$ and $A_C$ are the sets of genes annotated to the parent and a child system, respectively. All parent systems with similarity $S(P, C) \geq 0.9$ with respect to one or more children were then removed, connecting all children of that system to all of its parents to maintain continuity of the hierarchical structure. After these operations, the resulting ontology had 9312 systems remaining (S1 Fig).

**Computing system principal components.**   For each system, we built a matrix **G** of dimensionality $N \times S$ representing all SNPs within 10 kb of any gene annotated to that system, where $N$ is the number of individuals in the cohort and $S$ is the number of SNPs annotated to that system, in arbitrary order. Each SNP matrix was projected onto its principal component (PC) decomposition, $\mathbf{T} = \mathbf{GW}$, using the technique of singular value decomposition, where **W** is of dimensionality $S \times k$ and represents the contribution of each SNP to each principal component. **T** is of dimensionality $N \times k$ and represents the projection for each of $N$ individuals in the $k$ dimensional PC space. The first $k$ PCs were kept, where $k$ is the number of PCs needed to explain 95% of the variance, limited to at most 50 PCs. The rationale for this dimensionality reduction was two-fold. Firstly, by discarding some of the variance, it accounts for LD between

SNPs and mitigates collinearity between predictors in the regression model. Secondly, it concentrates significance when testing large SNP matrices having many degrees of freedom. To reduce computation time for large data sets we also included the option to use randomized singular value decomposition [59] implemented in the *rSVD* R package [60]. This option drastically reduces the principal component computation at the cost of reduced accuracy.

## Statistical analysis

**Hierarchical association analysis.** We next computed the association of each system with phenotypic outcome using a generalized linear model [61,62]. For each system, we trained a model $f_1$: $g(\mathbf{y}) = \mathbf{T}\boldsymbol{\alpha}+\mathbf{C}\boldsymbol{\beta}$ +$\mathbf{s}$, where $\mathbf{y}$ is the vector of phenotypic outcomes for each of the $N$ individuals, $\mathbf{T}$ is the principal component matrix defined above, $\mathbf{C}$ is a matrix holding an $N \times c$ set of clinical covariates, $\boldsymbol{\alpha}$ is a vector of genetic effects for each of the $k$ system PCs, $\boldsymbol{\beta}$ is a vector of effects for the $c$ covariates, $\mathbf{s}$ is the vector of residuals for each of the $N$ individuals, and $g$ is the link function, chosen to be logit to model a binary case/control phenotype. This model was compared to the null model $f_0$: $g(\mathbf{y}) = \mathbf{C}\boldsymbol{\beta} + \mathbf{s}$ using a logistic regression likelihood ratio ($\chi^2$) test. $\mathbf{C}$ included covariates of age, sex, current smoking status, and total pack-years, as well as the first 10 population principal components computed from the entire genome-wide set of variants to account for population structure. For each system determined to have significant genetic contribution (i.e. for which $f_1$ is significant), we decomposed $\boldsymbol{\alpha}$ as a SNP's effect vector $\mathbf{e} = \mathbf{W}\boldsymbol{\alpha}$, noting that the columns of $\mathbf{W}$ are the first $k$ eigenvectors of $TT$. The resulting vector $\mathbf{e}$ captures the total effect contributed by each SNP to this model while accounting for LD. It should be noted that, while this is the amount of total genetic effect captured by the PCs, it is not true of the total genetic effect captured by the SNPs in the gene set (since some PC that account for less than 95 percent of the variance are pruned away).

**Empirical assessment of significance via permutation testing.** The type I error rate of this association test was estimated empirically using permutation testing. For this procedure, we repeated the association test 10,000 times while randomly permuting the outcome variable while balancing the clinical covariates as described in [63] to generate a null distribution with no disease-associated SNPs. The type I error rate was taken as the mean fraction of systems with $p < 0.05$ over all permutations. This mean was 0.052 ($\sigma = 0.00185$), suggesting the error rate was well-controlled. We also produced log-qqplots (S7 Fig) to demonstrate the permuted null's p-values adherence to a uniform distribution, while the actual cohort's p-value are inflated, suggesting a true signal.

In addition, to confirm that the specific gene sets identified by the above tests were significant relative to random gene sets of the same size and position in the ontology hierarchy, we generated 100 random ontologies in which gene-system assignments were randomly permuted while retaining the hierarchical structure (parent-child relationships between the systems). For each system reported as significant in the earlier analysis, we then performed the same Higana process (see Computing system principal components and Hierarchical association analysis, above) for each of the 100 permuted ontologies, resulting in 100 p-values of association. This null distribution of 100 p-values was then used to recalibrate the actual p-value of association obtained previously. This adjustment procedure thus implemented a "competitive" test, i.e. to reject the null hypothesis that genes of equal size and position in the hierarchy could produce the observed association [26].

**Nested association tests.** A common challenge with Gene Ontology-based analysis is the prevalence of very similar (nested or overlapping) gene sets reported as significant. Such analysis will often result in not just one significant gene set, but several highly related sets nested within the same subhierarchy, all of which reflect the same functional concept but at different

resolutions of generality/specificity. We also noted that a single gene containing a strongly associated variant might drive the significance of a gene set, even in the absence of convergent signal from other genes. To overcome these artifacts, we implemented two additional tests for identifying systems for which the signal is mainly driven by a child system or single gene:

1. <u>Top child test</u>: In this association test, we identified the most significant child system for each system under consideration and excluded SNPs near genes annotated to the top child.

2. <u>Top gene test</u>: We computed an association score for each gene by repeating the association where each gene set is a single gene (S2 Data). We then identified the most significant gene for each system and excluded all SNPs near that gene.

In cases where a SNP is near both a gene or system identified for removal and one that should be kept, we chose to be strict and always exclude the SNP regardless.

**Test by simulation.**   In order to test our method in a controlled scenario, we devised two simulation schemas. In both schemas, we simulated 10 binary SNPs associated with a phenotype for 1000 case samples and 1000 control samples, representing a system that is associated with a hypothetical phenotype. In the first schema, the SNPs were generated independently, but with a probability of 0.02 for case versus 0.01 for controls. For the second schema, in order to simulate correlation structure, SNPs were generated with an Ising model using the Ising-Sampler package in R, with a random connectivity graph with connectivity probability of 0.5 between all nodes, and a threshold parameter of -3.5 for case and -5 for control, resulting in more case SNPs than control SNPs [64]. To both of these simulated SNP probabilities, we added increasing ammounts of noise, swept between zero and 0.1. This whole process was repeated 100 times for statistical stability.

Once the SNPs were simulated, we tested two contrasting approaches; one- and two- step statistical analysis. One-step analysis is the approach taken by Higana, that is, PCA followed by a chi-squared analysis. Two-step analysis represents the competing approaches (PLINK and MAGMA) which derive per-SNP or per-gene summary statistic, and then use Fisher's method to combine the relevant statistics for a system of genes (S8 Fig). Power was calculated as 1-(the proportion of tests that failed to reject the null hypothesis with a p-value of 0.05).

## Supporting information

**S1 Fig. Distribution of size of systems in the hierarchy constructed from Gene Ontology.** (a) All systems. Horizontal axis is log-scaled. (b) Restricted to systems of up to 75 genes. (TIF)

**S2 Fig. Manhattan plot of case-control association analysis of COPDGene non-Hispanic White genotype data.** Horizontal line denotes $p = 5 \times 10^{-8}$. (TIF)

**S3 Fig. Conditional association test results for the muscle atrophy subhierarchy.** Bar heights indicate -log10(p-value). P-values are unadjusted. Comparison of association results for the original uncorrected analysis (red) and conditional tests (blue- "top child", green- "top gene"). Shaded bars for the "top child" test indicate the system had no children and thus no correction was applied. (TIF)

**S4 Fig. Comparison of significant systems returned by Higana, PLINK and MAGMA.** (a) Venn diagram of systems returned at strict FWER < 0.05 for all methods. (b) Venn diagram of systems returned with relaxed p < 0.001. (c-e) Log-scaled p-values for all tested systems.  is the

Pearson correlation coefficient between the log-scaled p-values of each pair of systems.
(TIF)

**S5 Fig. Subhierarchy of significantly associated systems from the non-Hispanic White cohort and more general ancestor Colored by p-value of association in the African American validation cohort.** System color corresponds to unadjusted p-value.
(TIF)

**S6 Fig. Number of distinct genes covered by the systems hierarchy when including systems up to a certain gene set size**
(TIF)

**S7 Fig. Higana Log QQ-plots** (a) The p-values of association of all terms in the COPD non-Hispanic white cohort, plotted against the expected uniform distribution. (b) The p-values of association of all terms from 10000 covariate balanced permutations, plotted against the expected uniform distribution.
(TIF)

**S8 Fig. Simulation results** a. Case or control SNPs were simulated using both an independent model or a correlated Ising model. Then these SNPs were passed to either one- and two-step evaluation procedures. b. Power calculation for simulations in the independent and c. Ising models
(TIF)

**S1 Data. Complete associations of terms with COPD in the non-Hispanic White cohort.**
(XLSX)

**S2 Data. Single Gene association with COPD in the non-Hispanic White cohort.**
(XLSX)

**S3 Data. MAGMA association of terms with COPD in the non-Hispanic White cohort.**
(XLSX)

**S4 Data. PLINK association of terms with COPD in the non-Hispanic White cohort.**
(XLSX)

**S1 Text.**
(DOCX)

## Author Contributions

**Conceptualization:** Daniel E. Carlin, Simon J. Larsen.

**Data curation:** Simon J. Larsen, Vikram Sirupurapu, Michael H. Cho, Edwin K. Silverman, Jan Baumbach.

**Formal analysis:** Daniel E. Carlin, Simon J. Larsen, Vikram Sirupurapu.

**Funding acquisition:** Edwin K. Silverman, Jan Baumbach, Trey Ideker.

**Investigation:** Daniel E. Carlin, Simon J. Larsen, Vikram Sirupurapu.

**Methodology:** Daniel E. Carlin, Simon J. Larsen, Jan Baumbach, Trey Ideker.

**Project administration:** Trey Ideker.

**Resources:** Michael H. Cho.

**Software:** Daniel E. Carlin, Simon J. Larsen, Vikram Sirupurapu.

**Supervision:** Jan Baumbach, Trey Ideker.

**Validation:** Daniel E. Carlin, Vikram Sirupurapu, Michael H. Cho.

**Visualization:** Daniel E. Carlin, Vikram Sirupurapu.

**Writing – original draft:** Daniel E. Carlin, Trey Ideker.

**Writing – review & editing:** Daniel E. Carlin, Simon J. Larsen, Michael H. Cho, Edwin K. Silverman, Jan Baumbach, Trey Ideker.

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
