## [Decision Letter · Decision Letter 0]

10 Aug 2021

PONE-D-21-16308

Hierarchical association of COPD to principal genetic components of biological systems

PLOS ONE

Dear Dr. Carlin,

Thank you for submitting your manuscript to PLOS ONE. After careful consideration, we feel that it has merit but does not fully meet PLOS ONE’s publication criteria as it currently stands. Therefore, we invite you to submit a revised version of the manuscript that addresses the points raised during the review process.

As you can see in the reviewers comments, there are some fundamental issues that need to be addressed before your work could be accepted for publication. Of those, the questions related to how the p-values are estimated is the most critical together with the need for additional work on simulated, in addition  to real, data.

We look forward to receiving your revised manuscript.

Kind regards,

Jordi Perez-Tur, PhD

Academic Editor

PLOS ONE

Journal Requirements:

2. Thank you for including the following ethics statement on the submission details page:

'The COPDGene study protocol was approved by the institutional review boards at the

21 participating clinical centers. All COPDGene participants provided written informed

consent.'

Please also include this information in the ethics statement in the Methods section of your manuscript.

TI is co-founder of Data4Cure, Inc., is on the Scientific Advisory Board, and has an equity interest. TI is on the Scientific Advisory Board of Ideaya BioSciences, Inc., has an equity interest, and receives sponsored research funding. The terms of these arrangements have been reviewed and approved by the University of California San Diego in accordance with its conflict of interest policies. MC is a consultant for Illumina and AstraZeneca. In the past three years, EKS has received institutional grant support from GSK and Bayer. 

5. Please note that in order to use the direct billing option the corresponding author must be affiliated with the chosen institute. Please either amend your manuscript to change the affiliation or corresponding author, or email us at plosone@plos.org with a request to remove this option.

6. Please remove your figures from within your manuscript file, leaving only the individual TIFF/EPS image files, uploaded separately.  These will be automatically included in the reviewers’ PDF.

Reviewers' comments:

Reviewer's Responses to Questions

**Comments to the Author**

1. Is the manuscript technically sound, and do the data support the conclusions?

Reviewer #1: Yes

Reviewer #2: Partly

2. Has the statistical analysis been performed appropriately and rigorously? 

Reviewer #1: Yes

Reviewer #2: No

3. Have the authors made all data underlying the findings in their manuscript fully available?

Reviewer #1: Yes

Reviewer #2: Yes

4. Is the manuscript presented in an intelligible fashion and written in standard English?

Reviewer #1: Yes

Reviewer #2: Yes

5. Review Comments to the Author

Reviewer #1: This study is an very interesting study using a new and experimental bioinformatic method to investigate the relationship between GWAS and COPD. In addition, they found novel results that has not been found in other studies to date.

Reviewer #2: How to incorporate pathway knowledge in genetic association studies has not been yet clear, and authors introduce a concise one-step approach, Hierarchical Genetic Analysis (Higana), which directly computes phenotype associations against each function in the large hierarchy of biological functions documented by the Gene Ontology. With the proposed method, authors identified risk genes and functions for COPD, highlighting microtubule transport, muscle adaptation, and nicotine receptor signaling pathways.

Major comments

1. P-value for the proposed method was based on the permutation. However if there are some correlations between covariates, the permutation-based p-value can be invalidated. For genetic studies, genotypes tend to be independent with other covariates such as age, and sex. However this is very important issue, and authors should provide the solution for this issue.

2. Related with the first comment, authors should provide results from the simulated data.

3. Based on Table 2, authors insisted that Higana and PLINK are much powerful than MAGMA. However it is based on the real data and results in Table 2 cannot be generalized to the general scenario. Empirical oower should be compared withe simulated data.

4. It is unclear how to handle the multiple testing problem.

5. I cannot find any link for R package or implementation about Higana. It is better to provide some code.

6. Permutation-based test is often very computationally intensive for GWAS. For instance, the significance level alpha= 5*10^-8 is often considered and too many replicates are necessary.

6. PLOS authors have the option to publish the peer review history of their article (what does this mean?). If published, this will include your full peer review and any attached files.

Reviewer #1: No

Reviewer #2: No

---

## [Author Response · Author response to Decision Letter 0]

15 Jun 2022

Reviewer #1: This study is an very interesting study using a new and experimental bioinformatic method to investigate the relationship between GWAS and COPD. In addition, they found novel results that has not been found in other studies to date.

Reviewer #2: How to incorporate pathway knowledge in genetic association studies has not been yet clear, and authors introduce a concise one-step approach, Hierarchical Genetic Analysis (Higana), which directly computes phenotype associations against each function in the large hierarchy of biological functions documented by the Gene Ontology. With the proposed method, authors identified risk genes and functions for COPD, highlighting microtubule transport, muscle adaptation, and nicotine receptor signaling pathways.  Major comments

 1. P-value for the proposed method was based on the permutation. However if there are some correlations between covariates, the permutation-based p-value can be invalidated. For genetic studies, genotypes tend to be independent with other covariates such as age, and sex. However this is very important issue, and authors should provide the solution for this issue.

The p-values for our methods are not solely based on permutation; in fact the primary association test is a chi-squared test that is not based in permutation at all. This test explicitly handles covariates, using a Chi-squared test to disprove the null hypothesis that the SNPs associated with a system are no more informative than the covariates. 

Permutation tests were only used for confirmation tests. In this case, we used permutations of the structure of the output labels only, which preserves (and therefore controls or) any correlations of covariates. 

 2. Related with the first comment, authors should provide results from the simulated data.

We have performed simulations with two simulation setups, with knowledge of true positive rates and increasing amounts of noise.

 3. Based on Table 2, authors insisted that Higana and PLINK are much powerful than MAGMA. However it is based on the real data and results in Table 2 cannot be generalized to the general scenario. Empirical power should be compared with the simulated data.

We have performed simulations and the empirical power curves now appear in the manuscript, as S7 Fig and supporting paragraphs.

 4. It is unclear how to handle the multiple testing problem.

In this case, multiple testing is not at the level of SNPs, but rather at the level of systems. As such, Bonferroni correction is applied for the number of systems, which is vastly less than the number of SNPs.

 5. I cannot find any link for R package or implementation about Higana. It is better to provide some code.

As mentioned in the manuscript, the code is available as an R package at http://github.com/SimonLarsen/higana/

 6. Permutation-based test is often very computationally intensive for GWAS. For instance, the significance level alpha= 5*10^-8 is often considered and too many replicates are necessary.

We agree with the reviewer that permutation of patient labels is not viable for the primary test of SNP association significance, for which the (alpha= 5*10^-8) figure pertains, and we avoid all tests of this sort in our method. In fact we use the parameterized chi-squared test as the primary test. However, we do perform two types of permutation tests to evaluate system, rather than SNP-wise, significance: the first is a permutation of the genes associated with each system, which establishes the significance of the system regardless of its position in the hierarchy. As this is a test of the system’s significance, the correct multiple test correction is the number of systems (9312). We also perform a permutation test of the patient labels, but only as confirmation of the system’s significance. Since only four test are necessary (one for each of the identified systems), multiple hypothesis correction is negligible.

---

## [Decision Letter · Decision Letter 1]

8 Aug 2022

PONE-D-21-16308R1Hierarchical association of COPD to principal genetic components of biological systemsPLOS ONE

Dear Dr. Carlin,

Thank you for submitting your manuscript to PLOS ONE. After careful consideration, we feel that it has merit but does not fully meet PLOS ONE’s publication criteria as it currently stands. Therefore, we invite you to submit a revised version of the manuscript that addresses the points raised during the review process.

We look forward to receiving your revised manuscript.

Kind regards,

Jordi Perez-Tur, PhD

Academic Editor

PLOS ONE

Additional Editor Comments (if provided):

Dear authors,

Thank you for submitting the revised version of your manuscript. It has improved from the previous version but some questions are still pending to receive a proper answer whereas some other arise. In the accompanying documentation, you'll find enclosed the comments from the reviewers. Please, answer them by modifying your manuscript or by providing an answer to the reviewer that raised each particular question.

With kind regards,

Reviewers' comments:

Reviewer's Responses to Questions

**Comments to the Author**

1. If the authors have adequately addressed your comments raised in a previous round of review and you feel that this manuscript is now acceptable for publication, you may indicate that here to bypass the “Comments to the Author” section, enter your conflict of interest statement in the “Confidential to Editor” section, and submit your "Accept" recommendation.

Reviewer #1: All comments have been addressed

Reviewer #2: (No Response)

2. Is the manuscript technically sound, and do the data support the conclusions?

Reviewer #1: Yes

Reviewer #2: Partly

3. Has the statistical analysis been performed appropriately and rigorously? 

Reviewer #1: Yes

Reviewer #2: Yes

4. Have the authors made all data underlying the findings in their manuscript fully available?

Reviewer #1: Yes

Reviewer #2: Yes

5. Is the manuscript presented in an intelligible fashion and written in standard English?

Reviewer #1: Yes

Reviewer #2: Yes

6. Review Comments to the Author

Reviewer #1: The responses has been addressed to reviewers comments well. I don't have any more questions and comments.

Reviewer #2: In this article, authors introduce Hierarchical Genetic Analysis (Higana), which directly computes phenotype associations against each function in the large hierarchy of biological functions documented by the Gene Ontology. Using this approach, they identify risk genes and functions for Chronic Obstructive Pulmonary Disease (COPD), highlighting microtubule transport, muscle adaptation, and nicotine receptor signaling pathways. The manuscript is written well. Here are my comments.

1. The p-value was obtained by permutation. However if the covariate adjustment is necessary, then standard permutation cannot be conducted. This problem becomes serious if covariates and SNPs are correlated. This must be taken into account for the permutation.

2. The authors did not provide QQ plot for Higana analyses, and it is not easy whether Higana controls the type-1 error correctly. It must be provided.

3. Gene-based test can be significant even when only one SNP is significant. Authors showed that 4 pathways are significantly associated with COPD but the p-value for each SNP must be checked.

4. Method is not clearly described, and authors should expalin their method clearly.

5. If I understand it correctly, all SNPs belonging genes for each pathway are utilized for PCA and PC scores are used for hierarchical association analyses.

If many SNPs are in linkage disequilibrium, and LD block sizes differ by genes, I think that such heterogeneity affcects the statiscal power of the proposed mothod. This problem must be addressed.

6. It is unclear why the proposed method is statistically more powerful compared to PLINK, and MAGMA. Any discussion is recommended why it performs better

7. PLOS authors have the option to publish the peer review history of their article (what does this mean?). If published, this will include your full peer review and any attached files.

Reviewer #1: No

Reviewer #2: No

---

## [Author Response · Author response to Decision Letter 1]

17 Apr 2023

Reviewer #1: The responses has been addressed to reviewers comments well. I don't have any more questions and comments.

Reviewer #2: In this article, authors introduce Hierarchical Genetic Analysis (Higana), which directly computes phenotype associations against each function in the large hierarchy of biological functions documented by the Gene Ontology. Using this approach, they identify risk genes and functions for Chronic Obstructive Pulmonary Disease (COPD), highlighting microtubule transport, muscle adaptation, and nicotine receptor signaling pathways. The manuscript is written well. Here are my comments.

1. The p-value was obtained by permutation. However if the covariate adjustment is necessary, then standard permutation cannot be conducted. This problem becomes serious if covariates and SNPs are correlated. This must be taken into account for the permutation.

There are two types of permutation tests applied here as noted in Empirical assessment of significance via permutation testing. The ontology permutation test does not disrupt the covariate-SNP relationships and therefore does not need this correction. However, we have now performed the standard label permutation balancing for clinical covarites as noted in the methods, which was used to evaluate the type 1 error rate of the test. The results here are similar to the previous result with the mean number of systems achieving a p-value of 0.05 under the balanced permutation being 0.052suggesting a good control of the type 1 error rate

2. The authors did not provide QQ plot for Higana analyses, and it is not easy whether Higana controls the type-1 error correctly. It must be provided.

We Have now included an additional supplemental figure 7 which provide the log qq plot for Higana and the null under balanced permutations as described above. This demonstrates both the inflation of p-values under the experimental condition, and the lack of deviation from the uniform distribution under the permuted null.

3. Gene-based test can be significant even when only one SNP is significant. Authors showed that 4 pathways are significantly associated with COPD but the p-value for each SNP must be checked.

We note that in fact we have checked the individual SNPs and genes for significance independently of the systems, and none of them are independently significant, as noted in the results paragraph 2. As an example, the Manhattan plot in Figure 3 shows all loci associated with GO:0047496 Vesicle Transport Along Microtubule highlighted in red, and none of them reach global significance.

4. Method is not clearly described, and authors should expalin their method clearly.

We have added some summary explaination in the results and explanation throughout methods to attempt to make our procedure more clear.

5. If I understand it correctly, all SNPs belonging genes for each pathway are utilized for PCA and PC scores are used for hierarchical association analyses.

If many SNPs are in linkage disequilibrium, and LD block sizes differ by genes, I think that such heterogeneity affcects the statiscal power of the proposed mothod. This problem must be addressed.

As we note in the discussion, the advantage of collapsing all SNPs genetic signal into PCs is that the variable number of causal SNPs, LD region sizes, and number of genes can all be reduced to a few number of summary variables, thereby providing more homogeneity to the statistical tests. We have added a statement in the discussion to emphasize this point.

6. It is unclear why the proposed method is statistically more powerful compared to PLINK, and MAGMA. Any discussion is recommended why it performs better

In paragraph 5 and 6 of the discussion we have sketch out why our method has an advantage over the other methods. These include the detection of multi-locus effects due to combination of SNP effects at a higher organizational level, the increased expressive power of combining effects the logistic regression, and the de-noising effect of PCA analysis.

---

## [Decision Letter · Decision Letter 2]

9 May 2023

Hierarchical association of COPD to principal genetic components of biological systems

PONE-D-21-16308R2

Dear Dr. Carlin,

We’re pleased to inform you that your manuscript has been judged scientifically suitable for publication and will be formally accepted for publication once it meets all outstanding technical requirements.

Kind regards,

Suda Parimala Ravindran, Ph.D.

Academic Editor

PLOS ONE

Additional Editor Comments (optional):

Reviewers' comments:

Reviewer's Responses to Questions

**Comments to the Author**

1. If the authors have adequately addressed your comments raised in a previous round of review and you feel that this manuscript is now acceptable for publication, you may indicate that here to bypass the “Comments to the Author” section, enter your conflict of interest statement in the “Confidential to Editor” section, and submit your "Accept" recommendation.

Reviewer #1: All comments have been addressed

Reviewer #2: All comments have been addressed

2. Is the manuscript technically sound, and do the data support the conclusions?

Reviewer #1: Yes

Reviewer #2: Yes

3. Has the statistical analysis been performed appropriately and rigorously? 

Reviewer #1: Yes

Reviewer #2: Yes

4. Have the authors made all data underlying the findings in their manuscript fully available?

Reviewer #1: Yes

Reviewer #2: Yes

5. Is the manuscript presented in an intelligible fashion and written in standard English?

Reviewer #1: Yes

Reviewer #2: Yes

6. Review Comments to the Author

Reviewer #1: (No Response)

Reviewer #2: Carlins and their colleagues carefully addressed my comments this time, and I do not have further comments.

7. PLOS authors have the option to publish the peer review history of their article (what does this mean?). If published, this will include your full peer review and any attached files.

Reviewer #1: No

Reviewer #2: No

---

## [Editor Report · Acceptance letter]

15 May 2023

PONE-D-21-16308R2 

Hierarchical association of COPD to principal genetic components of biological systems 

Dear Dr. Ideker:

I'm pleased to inform you that your manuscript has been deemed suitable for publication in PLOS ONE. Congratulations! Your manuscript is now with our production department. 

Kind regards, 

on behalf of

Dr. Suda Parimala Ravindran 

Academic Editor

PLOS ONE